# Low Divergence Among Natural Populations of *Cornus kousa* subsp. *chinensis* Revealed by ISSR Markers

**Jia-Qiu Yuan [1,2], Qin Fang [1,2], Guo-Hua Liu [3] and Xiang-Xiang Fu [1,2,*]**

1   Co-Innovation Center for Sustainable Forestry in Southern China, Nanjing Forestry University,
    Nanjing 210037, China; jq.yuan1001@gmail.com (J.-Q.Y.); qinfang1214@163.com (Q.F.)
2   College of Forestry, Nanjing Forestry University, Nanjing 210037, China
3   College of Forestry, Jiangsu Vocational College of Agriculture and Forestry, Jurong 212400, China;
    Liuguohua@jsafc.edu.cn
*   Correspondence: xxfu@njfu.edu.cn

**Abstract:** *Research Highlights:* Taking Chinese dogwood (*Cornus kousa* subsp. *chinensis*) as an example, the genetic characteristics of natural populations collected from main a distribution area were evaluated using intersimple sequence repeat (ISSR) markers to reveal the genetic basis for further selection and breeding. *Background and Objectives:* Chinese dogwood is a small understory tree that is widely distributed in China. Chinese dogwood has attracted interest for its potential horticultural and ornamental values, and its natural resource potential urgently needs to be estimated. *Materials and Methods:* In this study, the genetic diversity of 12 natural populations collected from six provinces containing 223 individuals was evaluated based on ISSR markers. *Results:* Relatively high levels of genetic diversity were found at both the population and individual levels. The Shannon's diversity index (I) among individuals (0.504) was higher than that among populations (0.338). Analysis of molecular variance (AMOVA) revealed that genetic variation mainly existed within populations (61.55%) rather than among populations (38.45%). According to the STRUCTURE analysis, 12 populations were assigned to two groups, i.e., the northern and southern ecological regions, which are separated by the Yangtze River. A Mantel test analysis showed that there was no significant correlation between genetic distance and geographic distance. *Conclusions:* Considering the breeding system of dogwoods, we speculated that the genetic characteristics of the natural populations of this species would be affected by the dispersal mode of its pollens and seeds; additionally, genetic drift could play an important role in its genetic differentiation. In conclusion, in situ conservation is recommended for Chinese dogwood based on our results.

**Keywords:** Chinese dogwood (*Cornus kousa* subsp. *chinensis*); ISSR; genetic diversity; genetic variation; population structure

---

## 1. Introduction

Trees in *Cornus*, belonging to the family Cornaceae, are a cluster of small evergreen or deciduous species with high ornamental value for their habit, showy bracts, and fruit that have good prospects as horticultural ornamental plants [1]. There are two groups in *Cornus*, the East Asian group and the North American group; unlike the North American group, the East Asian group grows beautiful edible aggregate fruit [2]. According to the classification system [3], the East Asian group of *Cornus* is divided into five species: *Cornus elliptica (Pojarkova)* Q. Y. Xiang & Boufford, *Cornus capitata* Wall, *Cornus kousa* Q. Y. Xiang, *Cornus multinervosa (Pojarkova)* Q. Y. Xiang, and *Cornus hongkongensis* Hemsley. In addition, *Cornus kousa* also contains two subspecies: *Cornus kousa* subsp. *chinensis* (Chinese dogwood) and *Cornus kousa* subsp. *kousa*. The East Asian group is distributed from the Himalayas to East Asia,

of which China is the main distribution area, including Inner Mongolia, Shaanxi, Gansu, Shanxi, Henan, and provinces south of the Yangtze River [4]. Abundant wild germplasm resources in *Cornus* have been found in China, especially for Chinese dogwood (*C. kousa* subsp. *chinensis*). Although the main distribution of its wild resources is in China, the value of *Cornus* as a potential ornamental plant has not yet been fully realized there [4]. Previous studies in dogwoods have mainly focused on population distribution, germplasm resource investigation, cultivation techniques, resistance, and seedling growth rhythm, as well as physiological and biochemical aspects [5–8]. Assessment of plant resources is the basis for further exploitation. As an extensively cultivated ornamental species, related evaluations in the North American group of *Cornus*, i.e., *Cornus florida* L. and *Cornus nuttallii*, have been documented [9,10]. However, studies on the East Asian group have not been performed. Thus, the diversity of the germplasm should be considered before the selection and breeding of ornamental cultivars. The optimal tool for assessing resources is currently molecular markers.

In recent years, DNA molecular markers including restriction fragment length polymorphisms (RFLPs), random amplified polymorphic DNA (RAPD), amplified fragment length polymorphisms (AFLPs), simple repeat sequences (SSRs), and intersimple repeat sequence interval polymorphism (ISSRs) have provided a reliable means of assessing genetic diversity [11]. As a dominant marker, ISSR is more reliable and repeatable, has a lower cost, and is more effective than other markers. It has been widely used for genetic diversity analysis in *Cornus officinalis* and *Cornus mas* L., and for the assessment of genetic stability in acclimated plantlets of *Cornus alba* L. [12–14]. To provide a theoretical basis for the conservation and utilization of *Cornus*, here, taking Chinese dogwood as an example, an evaluation will be carried out for germplasm across the distribution area based on ISSR markers.

## 2. Materials and Methods

### 2.1. Sample Collection

A total of 223 samples of Chinese dogwood were collected from 12 natural populations in six provinces in China (Figure 1). The latitude and longitude of the collection sites were recorded with a portable GPS (PokeNavi map21EX; Empex Instruments, Tokyo, Japan), and the number of samples collected for each population is shown in Table 1. The sampled individuals were adult trees that were randomly distributed in populations at a distance of more than 50 m from each other. The fresh leaves were collected and sealed in plastic bags filled with silica gel, and then stored at −20 °C for subsequent DNA extraction.

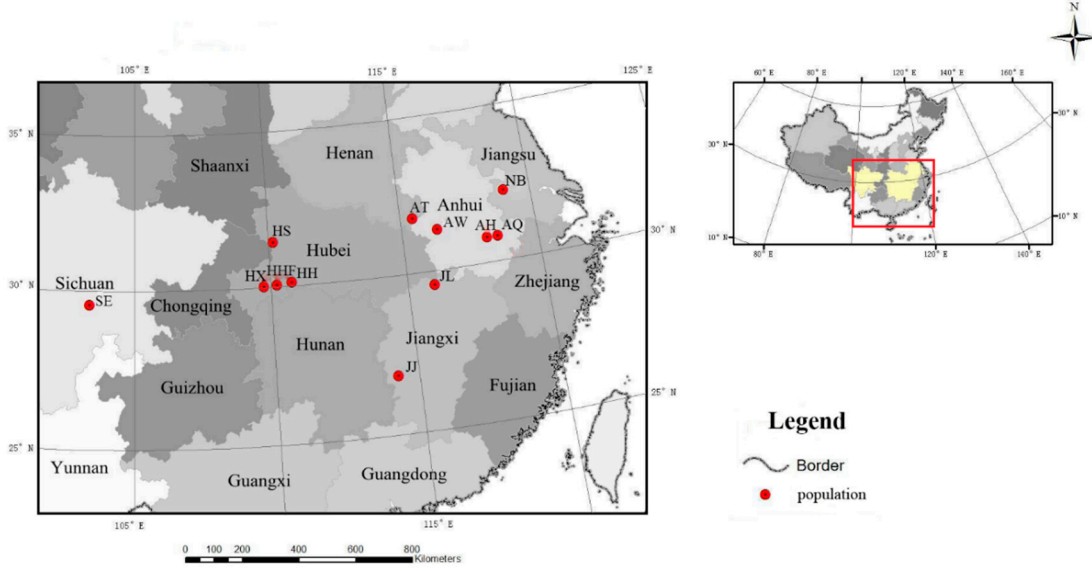

**Figure 1.** Distribution of the *Cornus kousa* subsp. *chinensis* sample in China.

**Table 1.** Information on *C. kousa* subsp. *chinensis* samples collected from natural populations.

| Population Code | Location | Number of Samples | Latitude (N) | Longitude (E) | Altitude (m) |
|---|---|---|---|---|---|
| HHF | Hefeng, Hubei | 25 | 30°03′29.27″ | 110°12′36.75″ | 1100~1526 |
| HX | Xuanen, Hubei | 14 | 30°01′52.07″ | 109°44′49.40″ | 1326~2016 |
| HS | Shennongjia, Hubei | 33 | 31°25′28.85″ | 110°10′17.44″ | 1100~2074 |
| AH | Huangshan, Anhui | 18 | 30°47′26.52″ | 118°6′19.87″ | 1162~1585 |
| AQ | Jixi, Qingliangfeng, Anhui | 17 | 30°47′40.20″ | 118°30′23.65″ | 700~1399 |
| AW | Shucheng, Wanfoshan, Anhui | 17 | 31°17′12.12″ | 116°19′59.59″ | 979~1400 |
| AT | Jinzhai, Tian Tang Zhai, Auhui | 22 | 31°44′42.00″ | 115°27′49.32″ | 990~1500 |
| SE | Leshan, Emei, Sichuan | 9 | 29°33′02.42″ | 103°21′45.21″ | 1120~1978 |
| HH | Shimen, Hupingshan, Hunan | 20 | 30°06′40.61″ | 110°46′44.27″ | 1100~2084 |
| NB | Nanjing, Baohua, Jiangsu | 8 | 32°07′45.54″ | 119°05′40.58″ | 200~300 |
| JJ | Ji'an, Jinggangshan, Jiangxi | 5 | 26°50′1.00″ | 114°15′42.00″ | 500~610 |
| JL | Jiujiang, Lushan, Jiangxi | 35 | 29°33′56.18″ | 115°57′36.22″ | 500~1350 |

## 2.2. DNA Extraction

The total DNA of samples was extracted from the leaves by a modified CTAB method [15] and then dissolved in sterile water for later analysis. The quantity and quality of the total DNA were assessed using 1% agarose gel electrophoresis and an ultraviolet spectrophotometer (Nanodrop 2000c; Thermo Scientific, America).

## 2.3. ISSR Analysis

After primary screening and rescreening for 100 ISSR primers (based on the ninth set of ISSR primers published by UBC) using an optimized amplification system and procedure, 10 primers were screened based on their high reproducibility and polymorphisms (Table 2). ISSR amplification was carried out in a modified 20-μL reaction volume consisting of 40 ng of DNA template, 1.0 μM of each primer, 8 μL 2 × Taq PCR MasterMix, and 8 μL ddH$_2$O. The PCR procedure in ProFlexTM BasePCR (ABI, USA) was as follows: predenaturation at 94 °C for 5 min, followed by 35 cycles of denaturation of the DNA at 94 °C for 30 s, primer annealing at 54–63 °C (depending on primer condition) for 45 s, primer extension at 72 °C for 1.5 min, and a final extension at 72 °C for 7 min. PCR products were separated by 2% TBE agarose gel stained with GelRed (BIOTIUM). Then, the gels were observed and photographed under an automatic digital gel image analysis system (Tanon 1600; Shanghai Tienergy Co., Ltd., China).

## 2.4. Data Analysis

Amplified fragments were scored for the presence (1) or absence (0) of homologous bands using Gelpro32 software (Media Cybernetics Inc., US) to read the image, and a 0–1 matrix diagram was formed. The Nei's genetic diversity (He), Shannon index or information index (I), percentage of polymorphic loci (PPL), total genetic diversity (H$_t$), intrapopulation genetic diversity (H$_s$), interpopulation genetic diversity (D$_{st}$) (D$_{st}$ = H$_t$-H$_s$), genetic differentiation coefficient (G$_{st}$) (G$_{st}$ = D$_{st}$/H$_t$), and gene flow (N$_m$) (N$_m$ = 0.5(1−G$_{st}$)/G$_{st}$) were also calculated using POPGENE32 [16]. Analysis of molecular variance (AMOVA) was performed to assess genetic diversity and inter/intrapopulation variability using GenAlEx v.6.5 [17,18]. A dendrogram tree was constructed from Nei's (1978) genetic similarity with the unweighted pair-group method of averages (UPGMA) using NTSys v.2.10 (NTSYS-PC 2.10, Applied Biostatics, Setauket, NY, USA). A principal component analysis (PCoA) was performed on Nei's genetic similarity matrix using NTSys v.2.10. A Mantel test was performed to analyze the correlation between the genetic distance and the geographical distance of the population using GenAlEx v.6.5 [17,18].

To examine the genetic structure, the model-based program STRUCTURE 2.2 [19] was used, with a burn-in period of 200,000 and with 200,000 Markov chain Monte Carlo iterations. Ten independent runs were performed setting the number of populations ($K$) from 1 to 12. The $K$ value was determined by the mean of estimated value of the Ln probability of data, called LnP (D), in the STRUCTURE output and an ad hoc statistic, $\Delta K$, based on the rate of change in the LnP (D) between successive $Ks$ [20].

## 3. Results

### 3.1. ISSR-Amplified Polymorphism

A total of 94 bands were obtained from 10 polymorphic primers, and the polymorphism ratio (PPL) was 100% (Table 2). The number of amplified bands was between six (UBC849) and 14 (UBC820) with an average of 9.4, and the band size ranged from 170 bp to 1500 bp. The results indicated that ISSR primers showed high polymorphism in Chinese dogwood.

**Table 2.** Amplification results for 10 intersimple sequence repeat (ISSR) primers in *C. kousa* subsp. *chinensis*.

| Primer | Sequences (5'~3') | Temperature (°C) | Total Bands | Polymorphic Bands | Polymorphism (PPL) (%) |
|---|---|---|---|---|---|
| UBC808 | AGAGAGAGAGAGAGAGC | 52 | 12 | 12 | 100 |
| UBC809 | AGAGAGAGAGAGAGAGG | 52 | 9 | 9 | 100 |
| UBC820 | GTGTGTGTGTGTGTGTC | 57 | 14 | 14 | 100 |
| UBC823 | TCTCTCTCTCTCTCTCC | 57 | 9 | 9 | 100 |
| UBC836 | AGAGAGAGAGAGAGAGYA | 52 | 10 | 10 | 100 |
| UBC842 | GAGAGAGAGAGAGAGAYG | 56 | 10 | 10 | 100 |
| UBC846 | CACACACACACACACART | 55 | 10 | 10 | 100 |
| UBC847 | CACACACACACACACARC | 57 | 7 | 7 | 100 |
| UBC849 | GTGTGTGTGTGTGTGTYA | 57 | 6 | 6 | 100 |
| UBC864 | ATGATGATGATGATGATG | 56 | 7 | 7 | 100 |
| Total | / | / | 94 | 94 | 100 |

### 3.2. Genetic Diversity Analysis

At the species level, the values of PPL, He, and I were 100%, 0.333, and 0.504, respectively. At the population level, the results are listed as follows: PPL ranged from 53.19% (Xuanen, Hubei (HX)) to 78.72% (Shimen, Hupingshan, Hunan (HH)), with an average of 66.31%; He ranged from 0.177 (Shucheng, Wanfoshan, Anhui (AW)) to 0.288 (Huangshan, Anhui (AH)), with an average of 0.226; and I varied from 0.274 (AW) to 0.423 (AH), with an average of 0.338 (Table 3). According to the values of He and I, the results showed that the genetic diversity of Chinese dogwood at the population level was slightly lower than that at the species level. The order of I of the 12 populations from high to low was: AH, Jixi, Qingliangfeng, Anhui (AQ), Hefeng, Hubei (HHF), Leshan, Emei, Sichuan (SE), HH, Shennongjia, Hubei (HS), Jinzhai, Tian Tang Zhai, Auhui (AT), Nanjing, Baohua, Jiangsu (NB), Jiujiang, Lushan, Jiangxi (JL), Ji'an, Jinggangshan, Jiangxi (JJ), HX, and AW. Furthermore, the highest genetic diversity among the populations, as revealed by the values of He and I, was in AH (0.288 and 0.423, respectively), while the lowest was in AW (0.177 and 0.274, respectively).

**Table 3.** Estimation of the genetic parameters for 12 natural populations of *C. kousa* subsp. *chinensis* based on ISSR markers.

| Population Code | Nei's Genetic Diversity (He) | Shannon Information Index (I) | Percentage of Polymorphic Loci (PPL, %) |
|---|---|---|---|
| HHF | 0.247 | 0.370 | 72.34 |
| HX | 0.190 | 0.283 | 56.38 |
| HS | 0.220 | 0.336 | 73.40 |
| AH | 0.288 | 0.423 | 75.53 |
| AQ | 0.272 | 0.401 | 72.34 |
| AW | 0.177 | 0.274 | 60.64 |
| AT | 0.221 | 0.331 | 65.96 |
| SE | 0.245 | 0.357 | 62.77 |
| HH | 0.229 | 0.353 | 78.72 |
| NB | 0.211 | 0.315 | 58.51 |
| JJ | 0.205 | 0.301 | 53.19 |
| JL | 0.199 | 0.304 | 65.96 |
| Population level | 0.226 | 0.338 | 66.31 |
| Species level | 0.333 | 0.504 | 100.00 |

## 3.3. Genetic Differentiation Analysis

The results of POPGENE32 showed that the values of $H_t$, $H_s$, $D_{st}$, and $G_{st}$ were 0.3396, 0.2253, 0.1143, and 0.3366, respectively. Gene differentiation among populations demonstrated that genetic diversity levels were low among populations (33.66%), and variations mainly emerged within populations (66.34%). In addition, the value of gene flow ($N_m$) among populations was higher, up to 0.9854, indicating that a certain level of gene flow could lead to small genetic differences among populations. Similarly, AMOVA analysis (Table 4) showed that the degree of genetic differentiation among the 12 populations was moderate, suggesting that 61.55% of the total genetic variation was within the population ($p < 0.001$), while 38.45% of the variation was among populations. This result further confirmed the low level of genetic diversity among the populations.

**Table 4.** Analysis of molecular variance (AMOVA) analysis for 12 natural populations of *C. kousa* subsp. *chinensis*.

| Source of Variation | df | Sum of Squares | Mean Squares | Variance Components | Percentage of Variance Components | Statistics | Value |
|---|---|---|---|---|---|---|---|
| Among pops | 11 | 1514.01 | 137.64 | 6.95 | 38.45% | | |
| within pops | 211 | 2348.52 | 11.13 | 11.13 | 61.55% | PhiPT | 0.3845 |
| Total | 222 | 3862.53 | 148.77 | 18.08 | 100.00% | | |

Note: PhiPT indicates the proportion of the total genetic variation among populations ($p < 0.001$).

## 3.4. Genetic Structure and Genetic Relationship Analysis

The genetic distance and genetic similarity in 12 natural populations of Chinese dogwood are listed in Table 5. The genetic distance ranged from 0.037 (between AQ and AH) to 0.297 (between JJ and AT), with an average of 0.177. Nei's coefficient, varying from 0.743 (between JJ and AT) to 0.964 (between AQ and AH), with an average of 0.840, indicated small differentiation among populations. According to the Mantel test, there was no significant correlation between genetic distance and geographical distance based on the ISSR data ($R^2 = 0.061$, $p = 0.08$), indicating that geographical isolation was not the main factor influencing the genetic differentiation of Chinese dogwood and that its genetic variation mainly exists within each population.

**Table 5.** Genetic similarity and genetic distance in 12 natural populations of *C. kousa* subsp. *chinensis*.

| Pop | HS | HH | HX | AH | AQ | JL | NB | AW | AT | HHF | JJ | SE |
|---|---|---|---|---|---|---|---|---|---|---|---|---|
| **HS** | *** | 0.928 | 0.915 | 0.886 | 0.896 | 0.866 | 0.890 | 0.867 | 0.863 | 0.858 | 0.785 | 0.819 |
| **HH** | 0.075 | *** | 0.960 | 0.887 | 0.887 | 0.816 | 0.869 | 0.896 | 0.879 | 0.851 | 0.791 | 0.817 |
| **HX** | 0.089 | 0.041 | *** | 0.891 | 0.879 | 0.790 | 0.849 | 0.843 | 0.848 | 0.861 | 0.780 | 0.824 |
| **AH** | 0.121 | 0.120 | 0.116 | *** | 0.964 | 0.823 | 0.866 | 0.821 | 0.842 | 0.835 | 0.763 | 0.796 |
| **AQ** | 0.109 | 0.120 | 0.129 | 0.037 | *** | 0.861 | 0.908 | 0.840 | 0.852 | 0.845 | 0.790 | 0.806 |
| **JL** | 0.144 | 0.204 | 0.235 | 0.195 | 0.150 | *** | 0.903 | 0.787 | 0.782 | 0.795 | 0.759 | 0.762 |
| **NB** | 0.116 | 0.140 | 0.164 | 0.144 | 0.097 | 0.102 | *** | 0.850 | 0.851 | 0.814 | 0.776 | 0.798 |
| **AW** | 0.143 | 0.110 | 0.171 | 0.198 | 0.175 | 0.239 | 0.162 | *** | 0.963 | 0.765 | 0.746 | 0.768 |
| **AT** | 0.147 | 0.129 | 0.165 | 0.173 | 0.161 | 0.246 | 0.162 | 0.038 | *** | 0.771 | 0.743 | 0.765 |
| **HHF** | 0.153 | 0.162 | 0.149 | 0.180 | 0.169 | 0.229 | 0.206 | 0.268 | 0.260 | *** | 0.849 | 0.877 |
| **JJ** | 0.242 | 0.235 | 0.248 | 0.271 | 0.235 | 0.276 | 0.254 | 0.294 | 0.297 | 0.164 | *** | 0.896 |
| **SE** | 0.200 | 0.202 | 0.193 | 0.228 | 0.216 | 0.272 | 0.226 | 0.264 | 0.268 | 0.131 | 0.110 | *** |

Note: the data above "***" represent the genetic similarity between the populations; the data below "***" represent the genetic distance (cM) between them.

A dendrogram was constructed according to the genetic similarity between the populations. Twelve populations were grouped into three clusters at a coefficient of 0.85 (Figure 2). The correlation coefficient (r) was 0.85, indicating that the clustering result was well. Cluster I, the largest, included seven populations (HS, HH, HX, AH, AQ, JL, and NB) mainly from Hubei, Anhui, Jiangsu, and Jiangxi provinces. The populations AW and AT from Anhui were placed in Cluster II. Cluster III, containing three populations (HHF, JJ, and SE), was mainly distributed in Hubei, Jiangxi, and Sichuan provinces. In addition, Cluster I and Cluster II can be combined into one larger group with a coefficient of 0.80. Additionally, the values of Nei's coefficient varying from 0.80 to 0.96 also indicate little differentiation among the groups (Figure 2). Among all individuals, three clusters were formed at a coefficient of 0.40 while the correlation coefficient (r) was 0.70. Individual samples in HS, HHF, and NB were grouped into cluster I, some samples of HH were placed to cluster II, and remaining samples were grouped into cluster III (SM 1). On the contrary, greater differentiation among the individual was verified by the values of Nei's coefficient varying from 0.34 to 0.96.

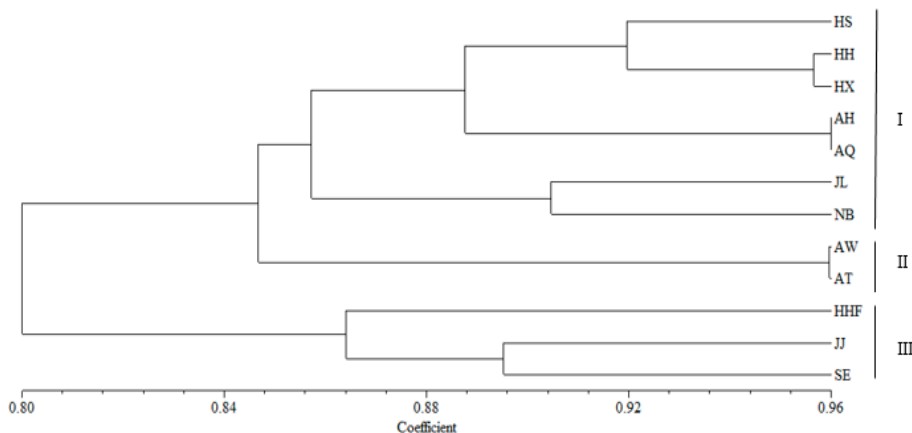

**Figure 2.** Unweighted pair-group method of averages (UPGMA) clustering of 12 populations of *C. kousa* subsp. *chinensis* based on ISSR markers.

Principal coordinate analysis (PCoA) can be used to analyze the phylogenetic relationships between species at different levels more intuitively than with systematic cluster analysis. The total percentage of variation explained by the first three axes was 62.78%, and it also showed that 12 populations could be divided into three groups at the 3D level (Figure 3). The three groups were as follows: the populations of HHF, SE, and JJ belonged to group I; the populations of HX, HH, HS, AH, AQ, JL, and NB belonged

to group II; and the populations of AT and AW belonged to group III. In addition, this result was consistent with the result of the system clustering with a coefficient of 0.85.

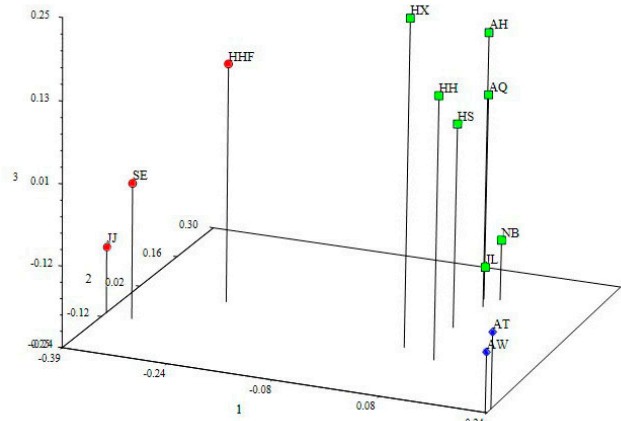

**Figure 3.** Principal component analysis (PCoA) analysis of 12 populations of *C. kousa* subsp. *chinensis* based on ISSR markers. Note: Red circles represent group I, green squares represent group II, and blue rhombuses represent group III.

The structure of a Chinese dogwood population containing 223 individuals was calculated based on 94 combined ISSR loci using a model-based approach in STRUCTURE. One hundred data sets were obtained by setting the number of possible clusters (*K*) from 1 to 12 with ten replications each. The LnP (D) value for each given *k* increased with increasing *k*, but no significant change was observed when *k* was increased from 1 to 12 (Figure 4A). The structure analysis showed that $\Delta K$ had a maximum value of 494.19 when *K* was 2, and $\Delta K$ for *K* = 3 ($\Delta K$ = 488.50) did not differ significantly from that for *K* = 2 (Figure 4B). Therefore, the 12 populations can be divided into two or three groups (Figure 4C). When *K* = 2, the model-based groups were divided into two groups (Figure 4C). Group I included the populations of HX, AH, AQ, HHF, SE, and JJ, and group II included the populations of HS, HH, NB, JL, AT, and AW (Table 6). When *K* = 3, the populations NB and JL were clustered into a group that was distributed on the edge of the Yangtze River (Figure 5). However, the model-based groups were not consistent with the isolated topographical distribution regions of the individuals. It also suggests that geographical isolation would not be the main factor responsible for genetic differentiation in Chinese dogwood. The three groups correspond to the three major germplasm sources in China, which was seven populations from three provinces in eastern China (Anhui, Jiangxi and Jiangsu), four populations from two provinces in central China (Hunan and Hubei), and one population from Sichuan province in southwestern China. Among them, group I was distributed in three major germplasm resource areas, while group II (except for HH) was concentrated north of the Yangtze River; the AT and AW populations were distributed in the Dabie Mountains (Figures 4C and 5).

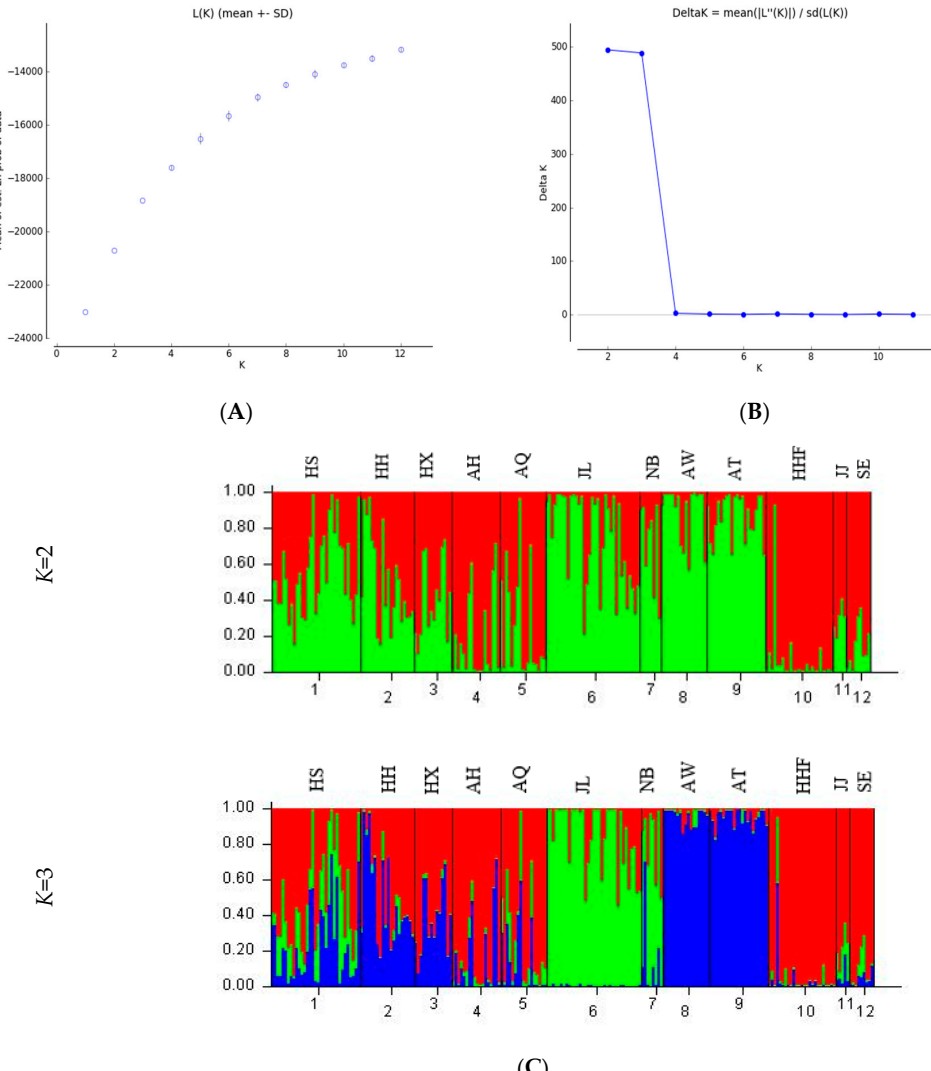

**Figure 4.** Population assignments based on STRUCTURE analysis of the ISSR marker for *C. kousa* subsp. *Chinensis.* (**A**) *K* and mean of the Ln probability of data, LnP(D); (**B**) ΔK broken-line graph with *K*; (**C**) the genetic clustering graph under the possible *K* values (*K* = 2,3).

**Table 6.** Proportion of membership for populations (locations) in three or four inferred subpopulations (*K* = 2,3) formed with 94 polymorphic ISSR markers.

| Population | *K* = 2 | | *K* = 3 | | |
|---|---|---|---|---|---|
| | 1 (red) | 2 (green) | 1 (red) | 2 (green) | 3 (blue) |
| HS | 0.4328 | **0.5672** | **0.5103** | 0.2539 | 0.2357 |
| HH | 0.4951 | **0.5049** | 0.4815 | 0.0351 | **0.4835** |
| HX | **0.5934** | 0.4066 | **0.6059** | 0.0140 | 0.3801 |
| AH | **0.8147** | 0.1853 | **0.8130** | 0.0342 | 0.1529 |
| AQ | **0.7381** | 0.2619 | **0.7306** | 0.1306 | 0.1388 |
| JL | 0.2451 | **0.7549** | 0.1415 | **0.8511** | 0.0074 |
| NB | 0.2873 | **0.7127** | 0.1911 | **0.6611** | 0.1477 |
| AW | 0.1040 | **0.8960** | 0.0260 | 0.0193 | **0.9547** |
| AT | 0.1301 | **0.8699** | 0.0362 | 0.0120 | **0.9518** |
| HHF | **0.9288** | 0.0712 | **0.9328** | 0.0300 | 0.0372 |
| JJ | **0.7018** | 0.2982 | **0.7782** | 0.1590 | 0.0628 |
| SE | **0.8489** | 0.1511 | **0.8998** | 0.0556 | 0.0446 |

Note: The highest proportions of membership for each *K* are in bold.

The result of the STRUCTURE analysis was not consistent with that of the UPGMA and PCoA analysis based on populations when *K* = 3, but was similar to that of the UPGMA analysis using samples at a coefficient of 0.52. The HHF, JJ, and SE populations were clustered in a group by UPGMA and PCoA analysis, while the proportion of membership for these populations was indeed larger than that of the others populations in group I according to the STRUCTURE analysis (Table 6). The STRUCTURE analysis showed that populations of JL and NB were grouped together and they deviated from their group in PCoA analysis. According to the results of AMOVA analysis based on three groups, the genetic variation among population was more significant (22.27%). Further analysis shows that the genetic backgrounds of three clusters have high similarity, and even some populations contain genetic characteristics of multiple groups.

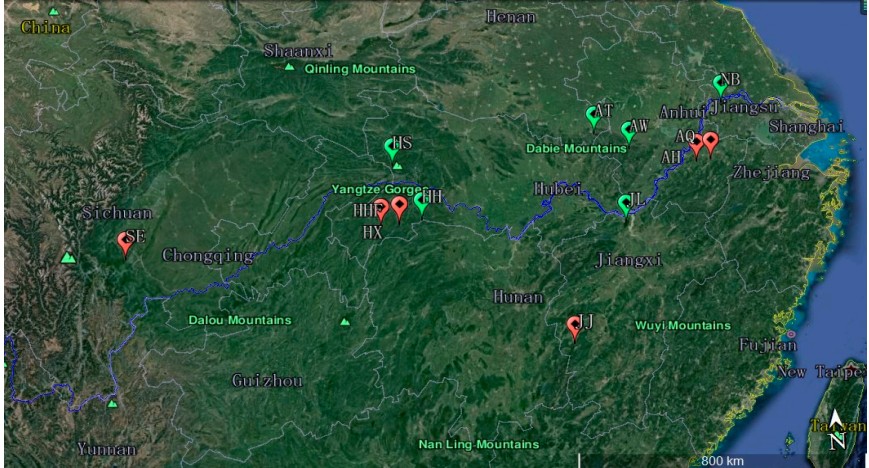

**Figure 5.** Geographic distributions of population structure in *C. kousa* subsp. *Chinensis.* Note: Subgroup I is in red and subgroup II is in green. The mountains are shown in green text. The Yangtze River is marked with blue lines.

## 4. Discussion

### 4.1. Genetic Diversity

Genetic diversity refers to the level of genetic diversity within a species and reflects the ability of a species to adapt to the environment [21]. In general, the indicators I and He are reliable indices for assessing the genetic diversity of a population. In this study, we found that the average values of He and I were 0.226 and 0.338 at the population level and 0.333 and 0.504 at the species level, respectively. Obviously, the higher genetic diversity occurred at the species level rather than at the population level for natural Chinese dogwood. Similar results have been reported in *C. officinalis* (the population level: 0.241, 0.361; the species level: 0.293, 0.446), *Magnolia officinalis* Rehd. et Wils. (P: 0.194,0.286; S: 0.342,0.496), *Davidia involucrata* Baill.(P: 0.220, 0.335; S: 0.330, 0.495), etc. [22–24]. However, obvious differences in genetic diversity among the 12 natural populations of Chinese dogwood were also revealed. For example, I and He in AH (0.423 and 0.288) were significantly higher than those in AW (0.274 and 0.177), though both populations come from Anhui province. Therefore, microclimate environments within the same region may result in differences in genetic diversity.

Plant genetic diversity is affected by many factors, including the evolutionary history, geographical distribution, and biological characteristics of a plant species [25]. Chinese dogwood has a long history, and diverse genes in its population were selected over time to allow it to adapt to heterogeneous habitats [3]. This is also supported by our results, in which the numerous variations in the 12 populations of Chinese dogwood occurred to match a wide range of environmental conditions. In terms of the breeding system of this species, its pollen is spread by insects (e.g., bees, butterflies), and its remote seeds dispersal depends on animals (especially birds); these factors create strong intergroup and

intragroup gene flow, which promotes gene exchange and leads to high genetic diversity [10,26]. Additionally, most samples were collected from nature reserves, including Huangshan, Qingliangfeng, Shennongjia, Lushan, and Emei, where forest community hierarchy and structure are preserved, there is less human disturbance, and less gene still remain. All these factors contribute to the abundant genetic diversity in populations of Chinese dogwood.

### 4.2. Genetic Differentiation and Genetic Structure

Remarkably, we detected a moderate level of population differentiation in Chinese dogwood ($G_{st}$ = 0.337), which was similar to that in *C. florida* (Cornaceae) ($G_{st}$ = 0.301) but slightly larger than those in *C. officinalis* ($G_{st}$ = 0.187) and *C. nuttallii* (Cornaceae) ($G_{st}$ = 0.090) [9,21,26]. A small genetic differentiation coefficient ($G_{st}$ < 0.25) suggests low genetic differentiation among populations, while a high value for gene flow ($N_m$ > 1) demonstrates a certain gene flow between populations and therefore weaker genetic differentiation [27]. For Chinese dogwood, gene flow ($N_m$ = 0.9854) was estimated by $G_{st}$, indicating that this level of migration would not prevent divergence among the populations. No significant correlation between the genetic distance and the geographic distance of this species was revealed by the Mantel test, which is consistent with the results for *C. florida* [26]. We speculated that the gene flow rate via seed dispersal was not high enough to counter the effect of genetic drift. This speculation is supported by Fischer et al. [28], who suggested that the lack of correlation between geographic distance and genetic distance resulted from genetic drift. In addition, the heterogeneous habitat of Chinese dogwood populations may also bring about random genetic drift that leads to genetic variation.

In general, the spatial distribution of genetic structure is closely related to the breeding mechanism of a species, reflecting ecological adaptive evolution, environmental changes, and natural selection effects [29]. Breeding systems and gene flow are the main factors affecting the genetic structure of populations [30]. Chinese dogwood could have a similar reproductive system as *C. florida*, which is a self-incompatible species [10,31], leading to a certain degree of nonselective mating and genetic variation. In addition, pollen and seeds are the two main vectors of genes and may also affect the differentiation of population structures in this study [32]. Most of the seeds are dispersed over long distances by migratory birds, weakening the differentiation among populations. However, Chinese dogwood is an entomophilous and animal-spread seed species, and its spreading range may be restricted by the range of the spreaders. The existing short-distance dispersal of seeds might partially explain the population structure evidenced in the AMOVA results (38.45% among populations; Table 4) [26]. It is inferred that this spatial genetic structure is mainly affected by the seed transmission mechanism, which is based on the dispersal trends of the fruit-eating birds [33].

### 4.3. Population Relationships

Twelve populations were assigned to three clusters based on the results of UPGMA and PCoA analyses, but no significant difference was observed among the coefficients of the clusters. We believed that the result of the STRUCTURE analysis (*K* = 3) was more valid because it further verified low differentiation among populations. Additionally, it occurred that different samples belonging to one population were clustered in different clusters in phylogenetic analysis using all samples, which may be caused by gene flow (SM 1). These results are in agreement with that suggesting low differentiation among the populations but greater differentiation within the populations by AMOVA analysis. However, we suspected that the main reason for some difference between the results of these clustering methods may be the ISSRs themselves. The population structure calculated by STRUCTURE (*K* = 2) showed that the 12 populations were divided into two groups: group I was distributed south of the Yangtze River (except the SE population), while group II was concentrated north of the Yangtze River (except the HH population). Geographically, two major ecological zones in the south and north divided by the Yangtze River were formed. In particular, the differentiation of HH from the central distribution is obviously small, while the SE (Sichuan Province) and NB (Jiangsu Province) populations,

from the fringe distributions in the west and east, respectively, are assigned to two groups, revealing the greater genetic distance between them (Figure 2). Based on the population structure of Chinese dogwood, it is supposed that the barrier between group II and group I would be water isolation. However, the low genetic differentiation among the existing populations was considered to correspond to a high frequency of gene flow and a continuous distribution. The distribution is similar to that of *Cyclocarya paliurus* (Batal.) Iljinsk, which has a continuous distribution [34], but quite different from that of *Liriodendron chinense* (Hemsl.) Sargent, which has a fragmented distribution [35].

### 4.4. Genetic Conservation and Germplasm Utilization Strategy

Generally, the protection of germplasm resources can be divided into in situ protection and ex situ protection. Ex situ conservation is the main and even the only conservation strategy for endangered species that have completely lost their habitat in the wild [36]. For species with high population differentiation and large genetic distances between populations, more populations are needed, while for species with low population differentiation and high genetic diversity within the population, fewer populations can meet the preservation needs of the species [37]. Considering the low genetic differentiation and high genetic diversity in populations of Chinese dogwood, in situ conservation is suitable for its cost-effective preservation of genetic diversity. Furthermore, the populations from the fringes of the distribution (including HHF, JJ, SE, NB, JL, AT, and AW) should be specially monitored in the future because these populations are likely to be genetically different from those in the center of the distribution area [38] and could contribute to genetic diversity and further development and utilization of the species.

## 5. Conclusions

In this study, we conclude that the lack of correlation between geographic distance and genetic distance resulted from genetic drift, especially that caused by heterogeneous habitats. Based on the dispersal mode of Chinese dogwood pollen and seeds, we speculate that the genetic structure is mainly influenced by seed transmission mechanisms, such as long-distance dispersal by migratory birds. Based on a cluster analysis, 12 populations were assigned to two groups, the northern and southern ecological regions, which are separated by the Yangtze River. Finally, due to the low genetic differentiation and high genetic diversity of Chinese dogwood, in situ conservation is suitable for the cost-effective preservation of the genetic diversity of this species.

**Author Contributions:** X.-X.F. and G.-H.L. developed the project. Q.F. provided DNA samples and expertise on the study system. J.-Q.Y. collected data, performed analyses, and wrote the paper.

**Funding:** This research was funded by Jiangsu Provincial Innovation and Promotion of Forestry Science and Technology (Project Number LYKJ [2018]06) and the Priority Academic Program Development of Jiangsu Higher Education Institutions (PAPD).

**Acknowledgments:** The authors also thank Qiang Lu and Xiao-Chun Li for their contributions in collecting samples.

**Conflicts of Interest:** The authors declare no conflict of interest.

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
