# Peer review of "Low Divergence Among Natural Populations of Cornus kousa subsp. chinensis Revealed by ISSR Markers"

_forests, doi:10.3390/f10121082_

Round 1
Reviewer 1 Report
The manuscript untitled “Low divergence among natural populations of Cornus kousa subsp. chinensis revealed by ISSR markers” by Yuan and colleagues address the genetic diversity and population struture of 12 natural populations from Cornus kousa subsp. chinensis, and evaluate its implications in conservation of genetic resources.
The manuscript is well-written and experiments are well conducted and analysed.
I believe that the paper would benefit with the inclusion of the state of the art of the studies already carried out in Cornus species using ISSR in the Introduction section. Moreover, in the discussion the data obtained should be compared with the ones published for other Cornus species, namely “Chinese Cornus officinalis: genetic resources, genetic diversity and core collection” Genet Resour Crop Evol (2012) 59:1659–1671.
Considering that the delta K from Structure analysis for K=3, did not differ significantly from K=2, it could be an added result to include the result along the K=2.

Reviewer 2 Report
The paper titled "Low divergence among natural populations of Cornus kousa subsp. chinensis revealed by ISSR markers" is focused to investigate the genetic diversity of 12 natural populations of Chinese dogwood collected in 6 provinces of China. The paper is potentially of interest, but the authors have to improve some points.
- First of all, in my opinion, the type of marker used (ISSRs are dominant and very low informative marker) is a limit. Why did the authors use ISSR markers? The authors didn’t show how these ISSRs amplify (some pictures as supplementary material need to be added)...the ISSR reproducibility is not (sometime) good and "clear". Wadl et al (2008; 2014) isolated a good panel of SSRs for Cornus mas L. and also (specific) for Kousa dogwood. Therefore, since microsatellites for the studied species are available, I don’t understand why the authors used ISSR markers. I suggest to genotype the samples by SSRs, in particular because the aim of the present paper (as reported from the authors) is highlight the “genetic basis for further selection and breeding”.
- How the authors selected 10 ISSR among “100 ISSR primer (based on the 9th set of ISSR primers published by UBC)”? It is not clear what (and how many) samples have been used to select 10 out of 100 markers.
- What does it mean Na and Ne (Table 3) for dominant markers? The analysis is based on dominant and multilocus markers, therefore Na and Ne for each population is not significant.
- The authors wrote (line 143-145) “..indicated small differentiation among populations”, but this result is not in agreement with dendrogram and PCoA analysis…explain? In my opinion is due to the markers used.
- The authors showed phylogenetic analysis for all population. But, since the authors underlined that the “variations mainly emerged within population”, it could be interestingly also verify and comment how the samples belonging to each population cluster together.
- In the cluster analysis, the authors have to show the bootstraps to verify the strength of analysis.
- Please use another visualization for PCoA analysis. In addition the authors have to highlight what is the total variability identified through this approach.
- In material and methods the authors wrote 233 samples and in structure 223…check!
- Did the authors are sure the "true" K value is 2? I suggest to increase the number of MCMC (also for a subset of K, e.g. 500,000 for K 2-6) and compare their results using also the Evanno’s method. The reference for STRUCTURE software is not correct (Stephens and Pritchard, 2003).
- The authors have to improve the discussion, also comparing their results to other studies developed in related species.
Author Response
We made corrections carefully following your comments and suggestions, and made explanation for some questions. In addition, we made language improving carefully. Hopefully, the revised MS can meet the approval of this journal. And don’t hesitate to let us know your any questions related to our MS.
The main corrections in revised MS and the response to the reviewer’s comments are as follows:
Reviewer #1
The manuscript untitled “Low divergence among natural populations of Cornus kousa subsp. chinensis revealed by ISSR markers” by Yuan and colleagues address the genetic diversity and population struture of 12 natural populations from Cornus kousa subsp. chinensis, and evaluate its implications in conservation of genetic resources.
The manuscript is well-written and experiments are well conducted and analysed.
The main issues regarding each of the sections in the paper are:
-Introduction (line 58-60):
Point 1: I believe that the paper would benefit with the inclusion of the state of the art of the studies already carried out in Cornus species using ISSR in the Introduction section.
Response 1: Thank you for your advice. We have supplemented it. Please see lines 58-60.
-Materials and Methods (line 80-81, 104-108):
Point 2: Which was the criteria for selecting the 10 primer sets from the total 100 ISSRs available?
Response 2: After primary screening and rescreening for 100 primers using optimized amplification system and procedure, 10 ISSR primers with high polymorphism and good repeatability were obtained for subsequent experiments.
Point 3: How to evaluate any spatial processes driving population structure?
Response 3: A continuous distribution that we may speculate, and the main reason was that group I was distributed in three major germplasm resource areas. This result requires further investigation and verification.
-Results (line 169-170, 192-198):
Point 4: The Figure could be improved, if coloring of symbols accordingly clusters identified.
Response 4: We have modified the figure. Please see lines 169-170.
Point 5: Considering that the delta K from Structure analysis for K=3, did not differ significantly from K=2, it could be an added result to include the result along the K=2.
Response 5: We agree with this point of view and have made changes. Please see lines 192-198.
-Discussion
Point 6: Moreover, in the discussion the data obtained should be compared with the ones published for other Cornus species, namely “Chinese Cornus officinalis: genetic resources, genetic diversity and core collection” Genet Resour Crop Evol (2012) 59:1659–1671.
Response 6: We agree with this point of view and add it to the discussion. Please see lines 222-224 and 242-244.
Thank you for your valuable and helpful advice again! We have also made corresponding modification in the latest MS. We would appreciate your comments.

Round 2
Reviewer 2 Report
The manuscript titled “Low divergence among natural populations of Cornus kousa subsp. chinensis revealed by ISSR markers” has been improved following some suggestions. Some gaps still remain “open” but the authors’ answers justified (more or less) them (e.g. the question about the type of marker used).
Anyway, the authors have to modify some points before publication. Following their answers, the authors have:
- POINT 3: to specify in the text how they chose 10 out of 100 available ISSR;
- POINT 4: on the contrary to what has been reported, the authors didn’t delete Na and Ne in their results.
- POINT 6: the authors wrote “And we can find this result intuitively based on phylogenetic analysis for all population. It occurred that different samples belonging to one population were clustered in different clusters, which was caused by gene flow”. In my opinion it is not sufficient, because they didn’t show results that supported their “insight”. Therefore, the authors have to add a phylogenetic analysis using samples (instead of populations) with some comments in the results/discussion, adding also the tree in supplementary material.
- Point 7: cophenetic correlation analysis is ok, but the authors have to add also this specification in the text and not only as answer.
- Point 8: as previous comment.
Author Response
The main corrections in revised MS and the response to the reviewer’s comments are as follows:
Point 3: to specify in the text how they chose 10 out of 100 available ISSR
Response 3: Thank you for your advice. We have supplemented it. Please see lines 81-83.
Point 4: on the contrary to what has been reported, the authors didn’t delete Na and Ne in their results.
Response 4: It is our mistake. Na and Ne have been deleted. Please check it in lines 95-96, 128-129.
Point 6: the authors wrote “And we can find this result intuitively based on phylogenetic analysis for all population. It occurred that different samples belonging to one population were clustered in different clusters, which was caused by gene flow”. In my opinion it is not sufficient, because they didn’t show results that supported their “insight”. Therefore, the authors have to add a phylogenetic analysis using samples (instead of populations) with some comments in the results/discussion, adding also the tree in supplementary material.
Response 6: We have added a phylogenetic analysis (SM 1) using samples and supplemented some comments in the results/discussion. Please see lines 161-167, 281-285.
Point 7: cophenetic correlation analysis is ok, but the authors have to add also this specification in the text and not only as answer.
Response 7: Thank you for your advice. We have supplemented it. Please see lines 155-157.
Point 8: as previous comment.
Response 8: We have highlighted what is the total variability identified through this approach. Please see lines 170-171.
Thank you for your valuable and helpful advice again! We have also made corresponding modification in the latest MS. We would appreciate your comments.